# Gaining an Understanding of Pneumocystosis in Wales

**DOI:** 10.3390/jof9060660

**Published:** 2023-06-13

**Authors:** Jonathan Ayling-Smith, Matthijs Backx, Elizabeth Grant, Rishi Dhillon, Jamie Duckers, Kerenza Hood, P. Lewis White

**Affiliations:** 1Cardiff and Vale University Health Board, University Hospital of Wales, Cardiff CF14 4XW, UK; 2College of Biomedical and Life Sciences, Cardiff University, Cardiff CF10 3AT, UK; hoodk1@cardiff.ac.uk; 3Public Health Wales, Cardiff CF10 4BZ, UK; 4All Wales Adult Cystic Fibrosis Centre, University Hospital Llandough, Penarth CF64 2XX, UK

**Keywords:** *Pneumocystis*, PcP, incidence, HIV, non-HIV, Wales

## Abstract

*Pneumocystis* pneumonia (PcP) is a serious complication of many significant immunocompromising conditions. Prior incidence estimates in Wales are based on PcP’s presentation in the HIV and transplant populations. The objectives were to describe the incidence of PcP in Wales using laboratory reporting measures and assess the impact of underlying immunosuppression cause on mortality. All positive PCR results for PcP between 2015 and 2018 were identified. The total number of unique positives with clinical and radiological correlation was 159 patients, a mean of 39.75 annually. The healthcare records of these patients were reviewed. The mortality at one month was 35.2% and 49.1% at one year. HIV remains the commonest cause of immunosuppression but has lower mortality than non-HIV conditions (12% vs. 59% at one year, *p* < 0.00001). Non-HIV conditions were categorised as life-threatening and non-life threatening but had a non-significant mortality (66% vs. 54%; *p* = 0.149), highlighting the negative impact of PcP. An incidence of PcP in Wales of 1.23–1.26 cases per 100,000 has been identified, 32–35% greater than the upper limit previously estimated. There is high mortality in non-HIV patients regardless of immunosuppression cause. A heightened awareness of PcP in these groups will hasten diagnosis and potentially improve mortality.

## 1. Introduction

*Pneumocystis* pneumonia (PcP) is a serious infection with a significant mortality rate [1]. Its ongoing importance as a global public health concern warranted its inclusion in the 2022 World Health Organisation fungal priority pathogen list [2]. Within the UK, the incidence of this condition is poorly understood due to a lack of active surveillance, the fact that fungal diseases, in general, are not notifiable, and the diagnosis of PcP is inconsistent. Estimated rates of PcP in the UK are 0.33–0.93 per 100,000 people [3]. The incidence of PcP in the established HIV-positive population has declined as a result of effective antiretroviral therapy and prophylaxis. However, it remains one of the leading causes of opportunistic infections among people with advanced HIV. Given that immunosuppressed patients are at higher risk of developing PcP and that this population is expanding annually through the use of novel immunomodulatory therapies [4], the incidence of PcP will vary according to local demographics and healthcare utility but has been shown to drive rates of PcP in national studies [5]. Patients with solid organ or bone marrow transplants and haematological cancers are well documented as being at particular risk, and the healthcare provision for this is concentrated in hubs in the UK. However, cases of PcP are increasingly documented in patients who are immunosuppressed secondary to treatment for a variety of rheumatological, renal, respiratory, and haematological disorders, which are likely to contribute to a significant burden of pathology [1,6]. At present, the true mortality risk to those undergoing treatment for these conditions is unknown.

While the burden of underlying diseases in Wales is broadly similar to other developed countries, unique clinical distinctions are evident. One in twelve adults report a longstanding respiratory condition, and in 2019, Wales had the highest avoidable mortality rate in Great Britain for respiratory diseases [7]. Wales has a lower proportion of solid organ transplants compared to other UK nations, with the majority being renal transplants [8]. The overall rates of HIV in Wales are low, with 2358 (2.4%) of the population of Wales receiving HIV care in 2019 [9]. However, in 2021, 42% of new cases of HIV in Wales were late diagnoses, generally presenting with AIDS-defining diseases, such as PcP [10,11]. The Welsh tertiary haematology service manages 140–160 acute myeloid patients and 30 acute lymphoblastic leukaemia patients and undertakes approximately 100 stem-cell transplants annually. The PcP diagnostic options across Wales are enhanced, with rapid access to both PcP PCR and serum (1–3)-β-D-glucan testing available for approximately 3.2 million people, which potentially provides information to accurately define an incidence of PcP in a large population. Microscopic detection of *P. jirovecii* is not routine practice in Wales; this approach has not been used for almost a decade due to its limited sensitivity and interpretative subjectivity.

To identify and understand existing and emerging service demands, assess patient risk, and improve diagnostic pathways, accurate knowledge of the incidence of PcP and associated mortality rates in different clinical cohorts is critical. Subsequently, this study describes attempts to both estimate the rates of PcP in Wales based on pre-established clinical risk but also correlates this data to laboratory-confirmed diagnoses of PcP based on PCR testing by the Public Health Wales Mycology Reference laboratory; it also investigates diagnostic accuracy and the impact of PcP on the patient outcome, depending on the underlying condition.

## 2. Materials and Methods

The data from a wider UK study stating a rate of 0.33–0.93 cases of PcP per 100,000 was applied to the population of Wales to predict the national number of PcP cases [3]. The accuracy of this estimate was corroborated through correlation with laboratory-diagnosed PcP cases, with the accuracy of diagnosis of each case determined by retrospective evaluation of clinical evidence. PcP PCR was performed using an in-house real-time PCR amplifying 77 bp of the mitochondrial 26S rDNA multi-copy gene with nucleic acid extracted using the BioMerieux EasyMag generic 2.0 protocol and PCR performed on the ABI 7500 real-time PCR platform, using oligonucleotides as previously described in [12], with technical procedures and clinical performance validated through ISO15189 accreditation, and analytical performance confirmed in a recent multicentre evaluation of PcP PCR methods and participation in external quality control schemes [13]. All positive PcP PCR-positive results from 2015 to 2018 were extracted from the Public Health Wales Mycology reference laboratory database. This laboratory processes samples from all hospital sites in Wales. This time period was chosen to avoid confounding from the COVID-19 pandemic and to ensure an adequate follow-up period when calculating rates of mortality within the population. All testing was performed as part of routine clinical diagnostics at the request of a consulting clinician, and this current study formed an audit of the clinical accuracy of this clinical service, not requiring ethical approval. All patients with more than one positive result were included only once. For patients with multiple positive results, the earliest PCR cycle number (quantification cycle, Cq) representing the highest fungal burden was included. One individual had multiple positive tests with greater than 12 months in between tests with negative tests in between, potentially indicating separate episodes of infection. This was included as two separate episodes of infection.

Based on meta-analyses and systemic reviews of the performance of PcP PCR testing of respiratory samples, PCR was assumed to be 95% sensitive and 90% specific for the diagnosis of PcP [14]. Subsequently, the total defined by laboratory testing was initially adjusted to reflect the 5% of missed cases missed and the 10% false-positivity rate to create a representative value independent of assay variability, which could be compared with estimates derived using the local assay.

Based on local clinical validation of the specific PcP PCR assay prior to the implementation into routine clinical use, all patients with a throat-swab PcP PCR-positive testing at <38 cycles were assumed to represent a true positive result, as locally, these results were highly specific (>98%). PcP PCR testing of upper respiratory tract samples has been associated with high specificities (≥96%) in other studies [15,16]. When testing deeper respiratory samples (bronchoalveolar lavage (BAL), non-directed bronchoalveolar lavage (NBL), or bronchial washings (BROW)), a Cq of <36 cycles was applied, which was again associated with high specificity (>95%).

All patient results with a positive PcP PCR result above the designated thresholds were individually reviewed, incorporating other laboratory results useful for the diagnosis of PcP (e.g., serum (1–3)-β-D-glucan, using the associate of Cape Cod Fungitell assay), radiology reports, as well as exploring clinical details gained from letters and discharge summaries. A decision was made on whether it represented a true clinical diagnosis based on the combination of results. Patients were considered not to have PcP if the combination of results did not represent a true clinical diagnosis (Figure 1). In line with international consensus definitions, all cases were classified as probable PcP [17].

All remaining patients were reviewed via their health records with their underlying diagnoses, the reason for immunosuppression and mortality recorded in an Excel database on a password-protected NHS network. The immunosuppression was clinically grouped for analytical purposes. Statistical analysis, including Chi-squared tests and 95% confidence intervals, were generated when comparing proportional values (e.g., the mortality between different groups) and *t*-test used for continuous variables. *p* values < 0.05 were considered to be statistically significant.

## 3. Results

When applying the UK-wide estimate of 0.33–0.93 cases of PcP per 100,000 to the population of Wales, an estimated 3,170,000 people, between 10 and 29 cases of PcP, would be expected annually in Wales [19]. Across the 4-year period (2015–2018), this would equate to 40–116 cases. However, a total of 289 PcP PCR-positive results were derived from the laboratory data, although this included 94 duplicate results, leaving 195 unique records. Of these, 103 (52.6%, 95% CI: 45.6–59.4) patients had PCR results associated with high assay specificity (>95%) and were considered true positives. The remaining 92 records were individually clinically reviewed, and 56 patients were considered to be clinically positive, generating a total of 159 cases (mean age 57, 66% male), indicating that 81.5% (95% CI: 75.5–86.6) of positive PcP PCR results were associated with actual disease; 73% (95% CI: 64–80%) of PcP PCR-positive BAL/NBL/Bronchial Wash compared to 94% (95% CI: 86–97) of upper respiratory tract samples (Figure 2).

If sensitivities and specificities from meta-analyses of PcP PCR are applied, there would be a minimal 5% reduction in cases resulting in 185 patients with PcP, supporting our initial process for defining PcP based on Ct value and combination with other fungal tests and clinical presentation (Figure 1). The 159 laboratory-positive cases equate to a mean number of 39.75 cases annually, but the study period is not a long enough period to determine a trend in incidence but provides an annual incidence of PcP in Wales of 1.23–1.26 cases per 100,000 and 32–35% greater than the upper limit estimated in the previous UK study [3] (Figure 3).

Fifty-three percent of samples involved bronchoscopy (58 broncho-alveolar lavage fluid, 25 bronchial washings, and 1 non-directed bronchial lavage fluid). Forty-two percent of the samples were throat swabs. The median cycle threshold for positivity in the throat swab samples was 37 (range: 26–41). The median cycle threshold for PCR positivity in BAL was 32 cycles (range: 20–41), which is significantly different to the throat swab (*p* < 0.0001) (Table 1).

The difference in median Cq does not translate to a difference in mortality either at 1 month or 1 year and likely reflects the greater fungal burden in the deeper respiratory samples compared to throat swabs. The mortality at 1 month of patients diagnosed by throat swab and BAL was 29% and 36%, respectively (*p* = 0.378). Mortality at one year was also not significantly different between sample types (42% vs. 48%; *p* = 0.514).

Of the 159 patients considered to be cases of PcP, mortality at one month was 35.2% (56/159, 95% CI: 28.2–42.9). One-year mortality was 49.1% (78/159, 95% CI: 41.4–56.8). Mortality between those that were considered definitively PCR-positive (Cq below the designated threshold) and those with later Cq value but with a clinical diagnosis of PcP were not statistically different at one month (36% and 34%, respectively, where *p* = 0.802) or one year (49% and 50%, respectively, where *p* = 0.861).

Regression analysis was undertaken to determine if there is a relationship between Cq value and mortality. In those that were HIV-positive, mortality was low regardless of Cq value and sample type. In those that were HIV-negative, when testing BAL/NBL or BROW (i.e, respiratory samples), the mortality rate for Cq < 37 was 45.7% (21/46, 95% CI: 32.1–59.8); for Cq ≥ 37, it was 35.0% (7/20, 95% CI: 18.1–56.7), which was not significantly different (*p* = 0.589). In the same HIV-negative cohort that had been tested using throat swabs, the mortality rate for Cq < 38 was 41.9% (13/31, 95% CI: 26.4–59.2); for Cq ≥ 38, it was 28.6% (6/21, 95% CI: 13.8–50.0), which was also not significantly different (*p* = 0.3887).

Underlying immunosuppressive conditions were grouped, as summarised in Table 2, and mortalities at one month and one year were calculated for groups with at least five participants to avoid bias associated with smaller cohorts and reduce identifiability.

The most common individual underlying condition associated with PcP remains HIV, with a total of 33 (21%, 95% CI: 15–28) patients diagnosed, but nowadays, almost four-fold more cases of PcP occur in HIV-negative patients (126 cases, 79%, 95% CI: 72–85). Mortality rates associated with HIV-positive PcP were low at one month (9%, 95% CI: 3–24) and one year (12%, 95% CI: 5–27). Thirty-one percent of the cases occurred in the heterogenous haematology/allogeneic stem-cell population, while the overall mortality at one month was 29% (95% CI: 18–42); mortality varied considerably depending on the underlying condition and clinical intervention. A significant burden of PcP (approximately 10% of cases) was associated with renal transplantation, rheumatological conditions, and solid cancer (Table 2). Compared to the HIV cohort, mortality was significantly increased in PcP patients with underlying rheumatological conditions, respiratory disease, and solid organ cancer. Rheumatology patients with PcP had the highest mortality rate within one month of infection, followed by patients with PcP and an underlying respiratory condition. Patients with PcP receiving treatment for solid organ cancer represented the group with the third highest mortality rate within one month of PcP infection, 47% (7/15, 95% CI: 25–70), and the highest mortality at one year, 80% (12/15, 95% CI: 55–93), both being significantly greater than mortality associated with PcP in HIV (one-month *P*: 0.0059; one-year *p* < 0.0001). The solid organ cancers group (*n* = 15) represented a heterogenous group of cell types/original organ disease, with lung cancer being the most common (*n* = 4). Clinical information regarding the modality of chemotherapy in these patients was limited, preventing further analysis. The latest one-year survival statistics by cancer type for Wales was used to identify the average percentage survival, weighted by the number of this population, with each solid organ cancer [20,21]. The average mortality at one year was, therefore, expected to be 36%, significantly different (*p* < 0.01) compared to the measured cumulative mortality of 80% associated with PcP in this cohort.

Subgroup analysis comparing “acute life-threatening” conditions with a mean life expectancy of 12 months or less with “non-acute or non-life-threatening” conditions demonstrated statistically significantly higher mortality for patients with “life-threatening conditions” and a PcP diagnosis compared to other conditions (66%, 33/50, 95% CI: 52–78 vs. 41%, 45/109, 95% CI: 32–51, *p* = 0.004). The breakdown of the conditions described as “acute life-threatening” or not is described in Table 2.

It was felt that HIV infection represents a unique cohort. Patients with HIV are often younger (mean age of HIV patients 49 (SD: 16.9) vs. 59 (SD: 16.5) than the non-HIV group, (*p* < 0.001) and clinically do well post-swift treatment with antimicrobials and antiretrovirals, as represented by the significant difference between PcP mortality at one year in the HIV group (12%, 4/33, 95% CI: 5–27) and non-HIV group (59%, 75/128, 95% CI: 50–67) (*p* < 0.00001). As such, if the HIV group is removed from the non-acute/non-life-threatening arm of the subgroup analysis, the difference in mortality between the two groups becomes non-significant (66%, 33/50, 95% CI: 52–78 vs. 54%, 41/76, 95% CI: 43–65, *p* = 0.149), highlighting the negative impact of PcP in non-acute/non-life-threatening/non-HIV conditions.

HIV, solid organ transplantation and stem-cell transplants are groups that have a large evidence base behind their established PcP risk. Further subgroup analysis was undertaken to compare the one-month and one-year mortality in patients with these three well-established risks of PcP compared with all other patients, which defined conditions less associated with PcP. The new risk factor patients consistently had both a one-month (43%, 39/91, 95% CI: 33–53) and one-year (65%, 59/91, 95% CI: 55–74) mortality that was significantly higher than the well-documented group (25%, 17/68, 95% CI: 16–36 at one month, *p* ≤ 0.02 and 28%, 19/68, 95% CI: 19–40 at one year, *p* < 0.0001). This significant difference in mortality at one month, however, does not remain the case when patients with HIV are removed (40%, 14/35, 95% CI: 26–56 vs. 43%, 39/91, 95% CI: 33–53 at one month, *p* = 0.771), but this difference returns again at one year (43%, 15/35, 95% CI: 28–59 vs. 65%, 59/91, 95% CI: 55–74, *p* = 0.02).

## 4. Discussion

Rates of PcP in Wales, assuming a high level of diagnostic accuracy in this study (1.23–1.26 per 100,000) are significantly higher than in previous UK-wide estimations (0.33–0.93 per 100,000), likely because the cases of PcP in immunosuppressed patients outside those cohorts with well-documented risk of PcP (e.g., solid-organ transplant and haematological malignancy) were not accounted for in previous studies [3]. The overall mortality of PcP in this study is 49.1% at one year, being 57.6% in non-HIV patients. Mortality remains high (54%) in non-HIV cohorts with conditions that do not pose an acute, significant threat to life at one year. This represents a group that is at a considerable risk of PcP but is possibly being under-investigated and likely subsequently undertreated, therefore, contributing to increased morbidity and mortality. While the literature describes the risks and strategies associated with more usual conditions, such as transplantation, there is a paucity of information regarding these other patients. However, their mortality post-PcP infection is consistently in excess of those cohorts where the risk of PcP is well-established. There are currently calls for clinical guidance for the diagnosis and management of PcP in rheumatology patients, with rheumatologists recognising the increased risk of PcP in their patients associated with prolonged use of high-dose corticosteroids but also biologic therapies [22]. The increased use of immuno-suppressive but also immunomodulatory therapies (e.g., infliximab) is leading to an expanding patient population potentially at risk of PcP, but currently not necessarily recognised as such. This is more problematic as there are likely to be a number of other patients in these cohorts who have died without the relevant investigations, as the index of suspicion was low. Indeed, 80% of PcP cases with underlying solid organ cancer died within one year, of which 47% died within a month of PcP diagnosis. The Wales national database for these specific cancers suggested that there was an expectation of a 68% mortality at 12 months, but the numbers of patients in this study are too small for comparison. The mortality of rheumatology conditions that were implicated in this work is expected to be low (<5%), but their mortality post-PcP is high (60% at 1 month and 66% at 1 year) [23]. It is not possible to compare to a rheumatology non-PcP cohort due to the multifactorial nature of these conditions and PcP’s relative rarity.

In the subgroup analysis, the risk of death is not significantly different whether there is a life-threatening condition vs. a non-life-threatening condition underpinning the immunosuppression and PcP infection. It implies that PcP causes a number of avoidable deaths in patients who have a condition with a mean life expectancy of greater than 12 months, such as rheumatoid arthritis and psoriasis. However, the risk of PcP in this cohort overall remains low, and there is limited information as to the level of immunosuppression required in this cohort to put them at increased risk. It is, therefore, difficult to predict the need for prophylactic strategies in these cohorts, but it should lead to heightened awareness of the risk of PcP and the need for enhanced diagnostic surveillance when patients present with acute respiratory illness.

It is not possible to derive disease-specific prevalence or incidence rates from this data as this work lacks the denominator, but the nature of these previously under-recognized cohorts suggests that diagnostic pathways for these novel at-risk patients can be improved. This is evident in the renal transplantation group, where PcP outbreaks have led to heightened awareness for diagnostics and infection control measures in this now-recognised high-risk group. The subsequent low mortality rate is likely a product of heightened awareness and rapid interventions in these patients [24].

A limitation of this study is the necessity of removing replicated patients. In each patient, the lowest Cq value was used to represent them during that acute illness. In circumstances where a throat swab was positive with a high Cq value and the patient went on to have a confirmatory bronchoscopy, in which the BAL yielded a low Cq value PCR result, the throat swab would be removed in preference to the BAL. This inherently introduces bias to those patients that had both but has no impact on the conclusions drawn regarding the underlying condition. While the lack of a reference method, such as immunofluorescence microscopy, may have resulted in the over-diagnosis of PcP based on PCR, an independent meta-analysis of PcP PCR on respiratory specimens has confirmed its accuracy as a diagnostic test. Indeed, the use of microscopic diagnosis would likely lead to an under-representation of the PcP burden. Furthermore, in our study, weak PcP PCR-positive results were investigated to confirm that the diagnosis and mortality rates were similar when compared to cases where the PcP PCR result was indicative of a high burden. This study also lacks a denominator for most underlying conditions, and it is, therefore, not possible to derive cohort-specific incidence rates.

## 5. Conclusions

The PcP case rate in Wales is higher than previously estimated, and although the true rates of these conditions in their subgroups are not known, it highlights non-transplant and non-cancer as a significant contributor to both the caseload and the mortality of PcP in Wales. However, developing diagnostic-driven pathways in this large group of patients remains problematic due to its size and complexity. Nevertheless, through the dual use of serum BDG and PcP PCR on respiratory samples, PcP can be confidently excluded and diagnosed. PcP can be diagnosed using a variety of respiratory samples, particularly in conjunction with other modalities, such as radiology and serology, and every effort should be made to attain a diagnosis using these relatively accessible tests. Further work needs to build on this audit with a focus on quantifying the rates of PcP and outcomes for patients with non-life-threatening conditions and exploring the mechanisms underpinning these observational findings. A heightened awareness of PcP in these groups will reduce delays in diagnosis and potentially improve mortality.

## Figures and Tables

**Figure 1 jof-09-00660-f001:**
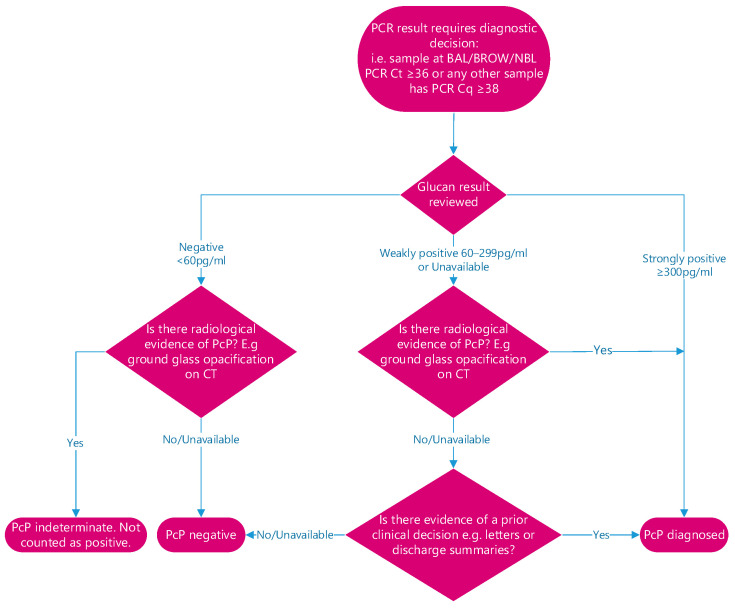
Diagnostic pathway for reviewing patients with a positive PcP PCR that does not reach established Cq cut-off values. When testing respiratory throat swabs, 126 out of 132 samples from patients without PcP generated a negative PcP PCR result (Specificity: 95.5%, 95% CI: 88.0–98.5. Positive likelihood ratio: 18.5). However, only two of these six false-positive results generated a cycle threshold (Ct value) < 38 cycles generating a specificity and positivity likelihood ratio of 98.5% (95% CI: 92.3–99.8) and 52.1, respectively, indicating that a PcP PCR-positive results with a Ct < 38 cycles on a throat swab was highly predictive of PcP. Similarly, when testing bronchoalveolar lavage fluid (*n* = 41), the specificity of PcP PCR when applying a positivity threshold of <36 cycles was 95.1% (39/41), and the positive likelihood ratio was 18.0. In relation to combination testing, involving reviewing patients who were deemed weakly positive by PcP PCR, the Fungitell (1–3)-β-D-Glucan threshold to exclude the need for clinical/radiological review was selected based on data from a recent study, where patients, who were PcP PCR-positive and had serum (1–3)-β-D-Glucan >200 pg/mL, the subsequent specificity of a PcP diagnosis was 100% [18]. In an attempt to minimise the impact of using different PcP PCR assays between studies and to improve our certainty of PcP diagnosis, it was decided to increase the (1–3)-β-D-Glucan positivity threshold to ≥300 pg/mL.

**Figure 2 jof-09-00660-f002:**
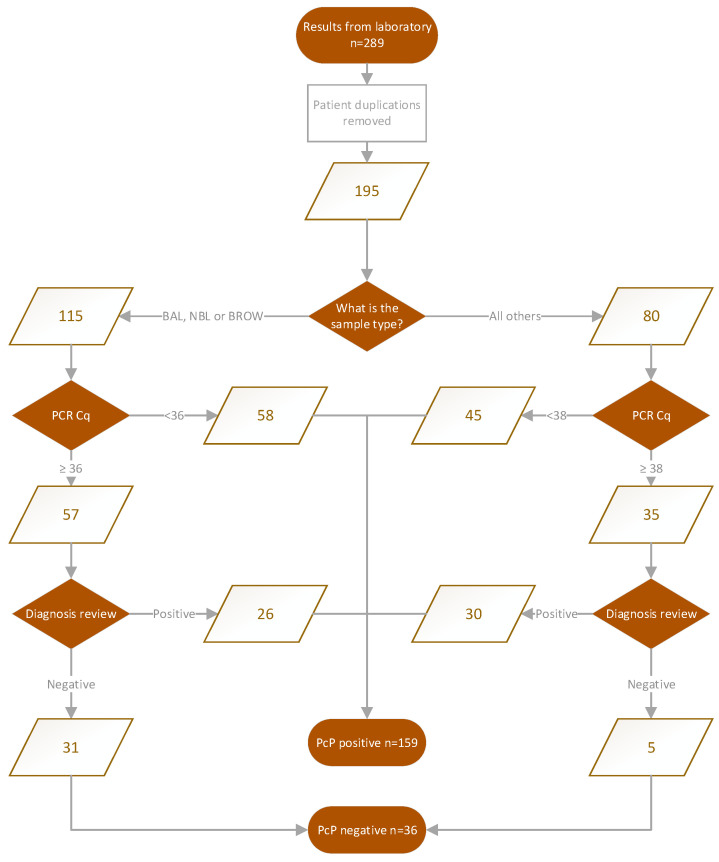
Flowchart of PcP PCR positivity and association with potential Pneumocystosis.

**Figure 3 jof-09-00660-f003:**
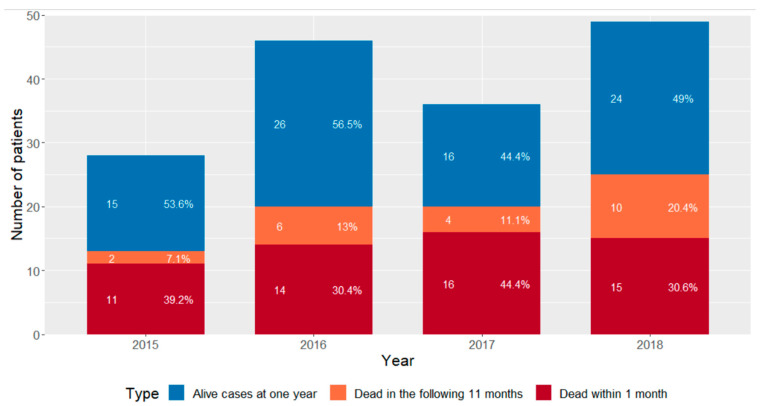
PcP mortality by year, separated into 1 month and 1 year.

**Table 1 jof-09-00660-t001:** PcP PCR Cq by site of sample.

Site of Sample	Number of Samples with PCR Cq < 36 in Deep Respiratory Samples and <38 in All Others	Number of Samples with PCR Cq ≥ 36 in Deep Respiratory Samples and ≥38 in Others	Number of Samples with PCR Cq ≥ 36 in Deep Respiratory Samples and ≥38 in Others in Keeping with Diagnosis of PcP	Total Number of Samples in Which PcP Diagnosed	Overall Median PCR Cq in PcP-Positive Patients	PCR Cq Range in PcP-Positive Patients
BAL	44	38	14	58	32	20–41
BROW	14	19	11	25	34	20–41
NBL	0	1	1	1	37	37
Throat swab	40	31	26	66	37	26–41
NPA	2	0	0	2	33.5	31–36
SNS	3	3	3	6	37	26–40
Serum	0	1	1	1	38	38
Total	103	93	56	159		

Sample sites include bronchoalveolar lavage (BAL), bronchial washing (BROW), non-directed bronchial lavage (NBL), throat swab, nasopharyngeal aspirate (NPA), sino-nasal secretions (SNS), and Serum.

**Table 2 jof-09-00660-t002:** Mortality of PcP by aetiology of immunosuppression.

Aetiology of Immunosuppression	Total Number	Mortality at 1 Month (95% CI)	Mortality at 1 Year (95% CI)	Acute Life Threatening?
HIV	33	9% (3–24%)	12% (5–27%)	No
Lymphoma	21	33% (17–55%)	62% (41–79%)	Yes
Renal +/− pancreatic transplant	17	35% (17–59%)	35% (17–59%)	No
Allogeneic stem-cell transplant	16	38% (18–61%)	44% (23–67%)	No
Rheumatology disease	15	60% (36–80%)	67% (42–85%)	No
Solid organ cancer	15	47% (25–70%)	80% (56–93%)	Yes
Haematology malignancy	12	8% (1–35%)	50% (25–75%)	Yes
Vasculitis	11	45% (21–72%)	64% (35–85%)	No
Respiratory disease	9	56% (26–81%)	67% (35–88%)	No
Inherited immunodeficiency	3	Population < 5	Population < 5	No
Autologous stem-cell transplant	3	Population < 5	Population < 5	No
Liver failure	3	Population < 5	Population < 5	Yes
Non-malignant renal disease	2	Population < 5	Population < 5	No
Autoimmune haematology disease	2	Population < 5	Population < 5	No
Inflammatory bowel disease	1	Population < 5	Population < 5	No
Dermatology disorder	1	Population < 5	Population < 5	No
Autoimmune neurological disease	1	Population < 5	Population < 5	No

Rheumatology disease, included rheumatoid arthritis, psoriatic disease, lupus, and other autoimmune diseases. Patients were included in the rheumatology, vasculitis, respiratory, autoimmune haematology, inflammatory bowel disease, dermatology, or autoimmune neurological disease only if they were on at least one of steroids, DMARD or biological therapy and not by the presence of condition alone. Patients were included in the solid organ cancer or haematological cancer groups if they were on treatment for that condition. Some patients had more than one cause for their immunosuppression and so belonged to more than one group; hence, a total is not applicable.

## Data Availability

Not applicable. As an NHS service evaluation, the data set cannot be made publicly available.

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
