# Peer review of "Gaining an Understanding of Pneumocystosis in Wales"

_jof, 2023, doi:10.3390/jof9060660_

Round 1

Reviewer 1 Report

I read with interest the study by Ayling-Smith et al. submitted to The Journal of Fungi (impact factor 5.724). The objectives were to provide original and additional data on Pneumocystosis in Wales.

The main results provided by this study are : “39.75 PCP cases annually; incidence of PCP in Wales 1;23 -1.26 cases per 100,000, 32-35 % greater than the upper limit estimated a previous study in United Kingdom.

Major comments.

Please explain why “gaining and understanding of PCP in Wales” / UK was necessary.

Are there specificities in Wales (population, ethnicity, geographical characteristics, health care system)?

Information on Welch population is needed (population, density)

Please indicate the hospital settings in which the study was performed

If I correctly understood, patients’ records were examined retrospectively. Retrospective patient inclusion was based on the results of P. jirovecii detection using a PCR assay.

All positive results were initially considered but the corresponding patients consisted of two groups.

First, patients with results of PCR in swab with < 38 cycles or patients with results of PCR in BAL or bronchial washes <36 cycles.

Second, patients with results of PCR in swab with ≥ 38 cycles or patients with results of PCR in BAL or bronchial washes ≥36 cycles. This second group was analyzed in the light of serum glucan dosages, radiological signs, final diagnosis considered by physicians …in order to exclude some patients without PCP

This approach is highly questionable.

1. please provide the genomic target of the PCR assay that was routinely used for PCP diagnosis

2. please provide criteria that have been used for the “local validation”; are the references 14 and 15 related to this PCR assay?

3. defining the cycle threshold values on the results of a meta-analysis (ref 14) that compared the results of diverse PCR assays (on diverse respiratory specimens, different genomic targets, different lab, different patient population) does not make sense.

4. what kind of glucan assay?

5. the threshold ≥300 pg per µL is also debatable since other invasive fungal infections can be associated with elevated glucan dosages; please explain how the thresholds 300, 60 to 299, and < 60 was chosen? To summarize: some PCP patients may have been overlooked; some patients without PCP may have been enrolled

6. Why did not you just consider the diagnosis listed in the patient records in order to include the patients in the study?

7. why did not you consider the results of microscopic detection of P. jirovecii organisms in BAL or bronchial washings?

Results

Table 1 “breakdown of PCR Cq by site of sample” in unnecessary. What “breakdown” mean, this term is not useful.

The paragraph on “subgroup analysis comparing “acute life threatening” vs. “non-acute non-life threatening” (as well as the paragraph in the discussion section) is unclear. Please rephrase and explain clearly what were “acute life threatening” and “non-acute non-life threatening” conditions.

The discussion section is long and verbose. It is unnecessary to re-write the introduction and the results…

Please focus on basis.

References

This section is a draft!

Ref 2. Please provide original link of WHO

Ref 6. Please provide journal name

Ref. 7. Is “Quality statement for respiratory disease” a reference? Is it a joke?

Ref. 8. Idem, see my aforementioned comments.

Ref 11. Please provide journal name or an accurate link

Ref 14. Please provide journal name

Ref 19. Please provide journal name

Minor comments.

The abbreviation "PCP PCR" is generally not used, it is "weird". Use PCR assay.

The style can be improved. Please write simple sentences starting with the subject with only one concept by sentence. Do not use metaphors.

At least, follow JOF guidance and provide complete affiliations and authors coordinates.

Supplementary materials: please provide a link, explain why the link was not indicated

Author Response

We thank the reviewers for taking the time and effort to review our manuscript and provide the useful feedback and insightful comments, which we have done our best to address as highlighted below in red text.

Reviewer 1

I read with interest the study by Ayling-Smith et al. submitted to The Journal of Fungi (impact factor 5.724). The objectives were to provide original and additional data on Pneumocystosis in Wales.

The main results provided by this study are : “39.75 PCP cases annually; incidence of PCP in Wales 1;23 -1.26 cases per 100,000, 32-35 % greater than the upper limit estimated a previous study in United Kingdom.

Major comments.

Please explain why “gaining and understanding of PCP in Wales” / UK was necessary.

This is addressed throughout the introduction with a more conclusive description See L50-71

Are there specificities in Wales (population, ethnicity, geographical characteristics, health care system)?

This is addressed in the introduction L50-60

Information on Welch population is needed (population, density)

See L60-62

Please indicate the hospital settings in which the study was performed

All hospital sites in Wales would be included in this study by the nature of using the Public Health Wales laboratory database. This laboratory processes all samples from Wales. No discrimination to hospital location i.e. outpatients, wards, critical care was given. This is highlighted in Methods in addition

 See L82-83

If I correctly understood, patients’ records were examined retrospectively. Retrospective patient inclusion was based on the results of P. jirovecii detection using a PCR assay.

All positive results were initially considered but the corresponding patients consisted of two groups.

First, patients with results of PCR in swab with < 38 cycles or patients with results of PCR in BAL or bronchial washes <36 cycles.

Second, patients with results of PCR in swab with ≥ 38 cycles or patients with results of PCR in BAL or bronchial washes ≥36 cycles. This second group was analyzed in the light of serum glucan dosages, radiological signs, final diagnosis considered by physicians …in order to exclude some patients without PCP

This approach is highly questionable.

  1. please provide the genomic target of the PCR assay that was routinely used for PCP diagnosis

Line 78

  1. please provide criteria that have been used for the “local validation”; are the references 14 and 15 related to this PCR assay?

Please see the amended legend to figure 1 which describes the specificity and likelihood of PcP according to Cycle threshold.

Reference 12 in the new manuscript reflects the analytical performance of the test as part of the international efforts to standardize PcP PCR.

  1. defining the cycle threshold values on the results of a meta-analysis (ref 14) that compared the results of diverse PCR assays (on diverse respiratory specimens, different genomic targets, different lab, different patient population) does not make sense.

We disagree, this approach was applied to confirm the accuracy of our estimates. If our assay generated significantly different performance to other PcP PCR tests then the our estimates could be misleading. By applying meta-analytical data that includes data generated by a range of different PCR assays but where statistical methods have been employed to generate generic performance parameters we can generate estimates of PcP to which we can compare our locally derived data.

We have clarified the purpose of this process (Lines 96-99). The estimate derived using this method resulted in a 5% reduction in case numbers, which we felt confirmed our estimates derived using the local PCR assay, and supported the algorithm used to define PcP (lines 149-153).

  1. what kind of glucan assay?

Line 110

  1. the threshold ≥300 pg per µL is also debatable since other invasive fungal infections can be associated with elevated glucan dosages; please explain how the thresholds 300, 60 to 299, and < 60 was chosen? To summarize: some PCP patients may have been overlooked; some patients without PCP may have been enrolled

We agree, but we not solely applying a positive BDG >300pg when defining PcP. These are patients who are also PCP PCR positive but with a PCR result at a Ct value where the specificity is <95%. Such a combination of BDG>300pg/ml with PcP PCR positivity would be highly indicative of PcP.

Please see the amended legend to figure 1 which justifies this threshold and reference 22 has also been included to support this decision

  1. Why did not you just consider the diagnosis listed in the patient records in order to include the patients in the study?

Many of these patients were diagnosed peri or post mortem. Their diagnoses are not routinely updated in patient records post mortem and death certification details were not available. Some positive results were obtained in Critical Care hour to days before the patient died. As such, it was appropriate to utilize this method of retrospectively analysing the data including radiology in order to determine if this represented a true positive. It also further emphasizes our conclusion that the pre-test likelihood is often felt to be low in some patients and so using their clinical records alone was felt to be inaccurate.

  1. why did not you consider the results of microscopic detection of P. jiroveciiorganisms in BAL or bronchial washings?

Microscopic detection of P.jirovecii is not routine practice in Wales. This approach has not been used for almost a decade due tot its limited sensitivity and subjectivity.

Results

Table 1 “breakdown of PCR Cq by site of sample” in unnecessary. What “breakdown” mean, this term is not useful.

We feel that this table is necessary as it allows for better data presentation of the results. It helps indicate that throat swab samples are associated with a high Cq value and BAL a lower value. It visually allows for a better understanding of the source of these results.

References to the word “breakdown” have been changed.

The paragraph on “subgroup analysis comparing “acute life threatening” vs. “non-acute non-life threatening” (as well as the paragraph in the discussion section) is unclear. Please rephrase and explain clearly what were “acute life threatening” and “non-acute non-life threatening” conditions.

This paragraph states that acute life threatening conditions are associated with a life expectancy of 12 months or less. The associated table clarifies which conditions are being discussed. The discussion paragraph has been altered to re-emphasize this.

The discussion section is long and verbose. It is unnecessary to re-write the introduction and the results…

Please focus on basis.

We feel the content of the discussion is justified but have amended the discussion in response to all the reviewers’ comments.

References

This section is a draft!

Ref 2. Please provide original link of WHO

Ref 6. Please provide journal name

Ref. 7. Is “Quality statement for respiratory disease” a reference? Is it a joke?

Ref. 8. Idem, see my aforementioned comments.

Ref 11. Please provide journal name or an accurate link

Ref 14. Please provide journal name

Ref 19. Please provide journal name

Apologies, there were issues with the referencing software which were not picked up. This section has been updated

Minor comments.

The abbreviation "PCP PCR" is generally not used, it is "weird". Use PCR assay.

We disagree, PcP PCR is a widely used and accepted term, and we prefer to continue using it.

The style can be improved. Please write simple sentences starting with the subject with only one concept by sentence. Do not use metaphors.

We feel this reflects differences in writing style and as this issue was not documented by the other reviewers, we prefer to maintain the current style when possible.

At least, follow JOF guidance and provide complete affiliations and authors coordinates.

This has been done. They were completed in full on the JoF website and not copied across correctly

Supplementary materials: please provide a link, explain why the link was not indicated

This was included in error, now deleted.

Reviewer 2 Report

This is an interesting review of PCP cases in Wales. It shows the increasing incidence of PCP in non-AIDS immunocompromised individuals. 

Minor issues:

Figure 1 and Table 1: check values. Should read < than 36  or 38.

Lines 219 - 220 seems a comma might be needed. 

Line 234: " a group that are... Please check grammar

Author Response

Reviewer 2

This is an interesting review of PCP cases in Wales. It shows the increasing incidence of PCP in non-AIDS immunocompromised individuals. 

Minor issues:

Figure 1 and Table 1: check values. Should read < than 36  or 38.

We have checked the references we have made to these figures in the methods and the local validation for <36 and <38 cycles. We feel that <36 and <38 cycles is an appropriate figure and therefore the inverse of this is ≥36 or ≥38. Figures 1 and 2 and table have been checked for consistency that all state <36/≥36 or <38/≥38. If your comment relates to a different issue which has not been addressed please let us know.

Comments on the Quality of English Language

Lines 219 - 220 seems a comma might be needed. 

This sentence contained an error that has been altered, making the comma now unnecessary.

Line 234: " a group that are... Please check grammar

Thank you this has been altered

Reviewer 3 Report

This manuscript presents an epidemiological review of recent Pneumocystis pneumonia cases in Wales including diagnostic methods and associated immunocompromising conditions. The study is timely and reports interesting findings in that incidence and outcomes are shifting due to the increasing use of immunomosuppressive therapeutic agents and the increasing efficacy of HIV diagnosis, antiretroviral therapy, and opportunistic pathogen prophylaxis and treatment. Accordingly, while HIV infection remains the most common predisposing condition, its relative incidence has decreased and its PcP-associated mortality is lower compared to other immunocompromising conditions. The manuscript is thoughtfully and well written, and the interpretations and conclusions are supported by the data.

Specific comment: In l 22, change “reduce delay to diagnosis” to “speed diagnosis.”

Author Response

Reviewer 3

This manuscript presents an epidemiological review of recent Pneumocystis pneumonia cases in Wales including diagnostic methods and associated immunocompromising conditions. The study is timely and reports interesting findings in that incidence and outcomes are shifting due to the increasing use of immunomosuppressive therapeutic agents and the increasing efficacy of HIV diagnosis, antiretroviral therapy, and opportunistic pathogen prophylaxis and treatment. Accordingly, while HIV infection remains the most common predisposing condition, its relative incidence has decreased and its PcP-associated mortality is lower compared to other immunocompromising conditions. The manuscript is thoughtfully and well written, and the interpretations and conclusions are supported by the data.

Specific comment: In l 22, change “reduce delay to diagnosis” to “speed diagnosis.”

We appreciate the comments above. In particular we recognise the need to remove ambiguity that may arise from using “reduce delay” as a potential double negative. However, we feel that speed diagnosis does not grammatically convey the message of the abstract and so have altered it to “hasten diagnosis”

Reviewer 4 Report

This paper retrospectively reviews, from laboratory reports, the incidence of Pneumocystis jirovecii pneumonia (PcP) in Wales between 2015-2018 and analyzes the impact of immunosuppression status as a cause of mortality within one month and one year of the episode. The reported incidence rate is 1.23-1.26 cases per 100,000 population; 32-35% greater than the upper limit previously estimated. The authors find that HIV infection remains the most frequent cause of PcP, but the mortality rate is lower than in non-HIV patients, regardless of the cause of immunosuppression. A higher alert for PcP in immunocompromised patients is encouraged in order to anticipate the diagnosis and reduce mortality.

INTRODUCTION

Pneumocystis pneumonia is a serious infection with a high mortality rate and has been included in the WHO list of priority fungal pathogens in 2022. The incidence of PcP in HIV patients is discussed, as well as the risk in immunocompromised patients secondary to treatment for various underlying diseases.

As indicated by the authors “this study describes attempts to both estimate the rates of PcP in Wales based on pre-established clinical risk but also correlates this data to laboratory confirmed diagnoses of PcP based on PCR testing by the Public Health Wales Mycology Reference laboratory, it also investigates diagnostic accuracy and the impact of PcP on patient outcome depending on the underlying condition”.

MATERIALS AND METHODS

Line 67: the reference number [2] seems erroneous. Do you mean ref [3]?

Figure 1 legend indicates “Diagnostic pathway of reviewing patients with positive PcP PCR”, but you only show cases where Cq values do not reach the established cut-off (36 or 38).

Figure 1: In the second column, we can read “Weakly positive/Unavailable 60-299pg/ml”. The word "Unavailable" does not seem to be in the right place; if so, please explain why.

Figure 1: An arrowhead on the horizontal "Yes" line leading from "Is there radiological....?" would help to understand the rationale.

L. 113 Please, specify the p values considered as statistically significant.

RESULTS

L. 138-142. These sentences refer to data of Table 1, please, mention this reference.

L. 143-162. In contrast, the authors compare mortality rates according to the type of sample and its Cq values, mentioning Table 1; however, no mortality data are shown in this table.

Table 2. It would be desirable to indicate as footnotes to the table the meaning of the acronyms used (some do not even appear in Material and Methods).

L. 167-174. These instructions look like footnotes to the table. This is probably an editing problem, but they should be separated from the main text.

L. 214-225. According to data in Table 2, the group of patients with HIV, solid organ transplantation and stem cell transplants amount to 66 cases (33+17+16), not 68 as reflected in the calculations for comparisons.

L.220. “…significantly lower higher than the well documented group…” , lower or higher, which one is correct?

DISCUSSION and CONCLUSIONS

The results of the analysis are compared with published data on similar categories of patients. In addition to the well-characterized groups, HIV, solid organ transplantation and stem cell transplants, the authors emphasize the increased mortality rates in other patients diagnosed with PcP who receive immunosuppressive treatments for other pathologies (rheumatologic diseases, vasculitis...).

A higher alert for PcP in immunocompromised patients is encouraged in order to anticipate the diagnosis and reduce mortality.

REFERENCES

Review the entire section. Some errors are listed below:

The species name “jirovecii” appears as “Jirovecii” in all references.

L.318-319: some data are repeated.

Ref. 6 introduces a private link, please, refer to the publication “Ann Pharmacother. 2016 Aug;50(8):673-9. doi: 10.1177/1060028016650107”

Ref. 7, 8 and 17 are not complete.

Ref. 19 is not complete. Year of publication? Curr Rheumatol Rep. 2017 Jun;19(6):35. doi: 10.1007/s11926-017-0664-6.

Author Response

Reviewer 4

This paper retrospectively reviews, from laboratory reports, the incidence of Pneumocystis jirovecii pneumonia (PcP) in Wales between 2015-2018 and analyzes the impact of immunosuppression status as a cause of mortality within one month and one year of the episode. The reported incidence rate is 1.23-1.26 cases per 100,000 population; 32-35% greater than the upper limit previously estimated. The authors find that HIV infection remains the most frequent cause of PcP, but the mortality rate is lower than in non-HIV patients, regardless of the cause of immunosuppression. A higher alert for PcP in immunocompromised patients is encouraged in order to anticipate the diagnosis and reduce mortality.

INTRODUCTION

Pneumocystis pneumonia is a serious infection with a high mortality rate and has been included in the WHO list of priority fungal pathogens in 2022. The incidence of PcP in HIV patients is discussed, as well as the risk in immunocompromised patients secondary to treatment for various underlying diseases.

As indicated by the authors “this study describes attempts to both estimate the rates of PcP in Wales based on pre-established clinical risk but also correlates this data to laboratory confirmed diagnoses of PcP based on PCR testing by the Public Health Wales Mycology Reference laboratory, it also investigates diagnostic accuracy and the impact of PcP on patient outcome depending on the underlying condition”.

MATERIALS AND METHODS

Line 67: the reference number [2] seems erroneous. Do you mean ref [3]?

Thankyou, yes this was in error, amended

Figure 1 legend indicates “Diagnostic pathway of reviewing patients with positive PcP PCR”, but you only show cases where Cq values do not reach the established cut-off (36 or 38).

Amended for clarity.

Figure 1: In the second column, we can read “Weakly positive/Unavailable 60-299pg/ml”. The word "Unavailable" does not seem to be in the right place; if so, please explain why.

“Unavailable” and “weakly positive” swapped position to make for better reading. New diagram needed to be uploaded but comment added to this effect

Figure 1: An arrowhead on the horizontal "Yes" line leading from "Is there radiological....?" would help to understand the rationale.

New diagram done to reflect this change

  1. 113 Please, specify the p values considered as statistically significant.

Added as stated

RESULTS

  1. 138-142. These sentences refer to data of Table 1, please, mention this reference.

First mention of table 1 reference moved to a more appropriate place in that paragraph

  1. 143-162. In contrast, the authors compare mortality rates according to the type of sample and its Cq values, mentioning Table 1; however, no mortality data are shown in this table.

Table 1 reference moved to a more appropriate location

Table 2. It would be desirable to indicate as footnotes to the table the meaning of the acronyms used (some do not even appear in Material and Methods).

Added as a footnote

  1. 167-174. These instructions look like footnotes to the table. This is probably an editing problem, but they should be separated from the main text.

This was an editing issue. Hopefully more readable now

  1. 214-225. According to data in Table 2, the group of patients with HIV, solid organ transplantation and stem cell transplants amount to 66 cases (33+17+16), not 68 as reflected in the calculations for comparisons.

We included autologous stem cell transplantation in this group as well as a cohort of patients with well documented/described clinical risk (+3). However, one patient had both HIV and a renal transplant so was not counted twice (-1).

L.220. “…significantly lower higher than the well documented group…” , lower or higher, which one is correct?

Thank you for noticing this. Higher is the correct word. Edited as such.

DISCUSSION and CONCLUSIONS

The results of the analysis are compared with published data on similar categories of patients. In addition to the well-characterized groups, HIV, solid organ transplantation and stem cell transplants, the authors emphasize the increased mortality rates in other patients diagnosed with PcP who receive immunosuppressive treatments for other pathologies (rheumatologic diseases, vasculitis...).

A higher alert for PcP in immunocompromised patients is encouraged in order to anticipate the diagnosis and reduce mortality.

We agree with the reviewers summary

REFERENCES

Review the entire section. Some errors are listed below:

The species name “jirovecii” appears as “Jirovecii” in all references.

L.318-319: some data are repeated.

Ref. 6 introduces a private link, please, refer to the publication “Ann Pharmacother. 2016 Aug;50(8):673-9. doi: 10.1177/1060028016650107”

Ref. 7, 8 and 17 are not complete.

Ref. 19 is not complete. Year of publication? Curr Rheumatol Rep. 2017 Jun;19(6):35. doi: 10.1007/s11926-017-0664-6.

Reference section reviewed and heavily edited. Please see response to reviewer 1.